# A model-based approach for detecting and identifying faults on the D.C. side of a P.V. system using electrical signatures from I-V characteristics

**Muhammad Adnan Khan[1]\*, Khalid Khan[2]\*, Adnan Daud Khan[3], Zubair Ahmad Khan[4], Shahbaz Khan[5,6], Abdullah Mohammed[7]**

1 Electrical Energy System Department, Center for Advanced Studies in Energy University of Engineering and Technology Peshawar, Peshawar, Pakistan, 2 Materials for Energy Storage and Conversion, Center for Advanced Studies in Energy University of Engineering and Technology Peshawar, Peshawar, Pakistan, 3 Renewable Energy, Center for Advanced Studies in Energy University of Engineering and Technology Peshawar, Peshawar, Pakistan, 4 Department of Mechatronics, University of Engineering and Technology Peshawar, Peshawar, Pakistan, 5 Capital University of Science and Technology (CUST), Islamabad, Pakistan, 6 Institute of Manufacturing Engineering Management, University of Engineering and Applied Sciences, Swat, Pakistan, 7 University Research Centre, Future University in Egypt, New Cairo, Egypt

\* adnankhan.muhd@gmail.com (MAK); Engrkhaliduet@gmail.com (KK)

**Data Availability Statement:** All relevant data are within the manuscript and its Supporting Information files.

## Abstract

With the development of distributed generation and the corresponding importance of the P. V. (photovoltaic) system, it is desired to operate a P.V. system efficiently and reliably. To ensure such an operation, a monitoring system is required to diagnose the health of the system. This paper aims to analyze a P.V. system under various operating conditions to identify parameters–derived from the I-V (current-voltage) characteristics of the P.V. system–that could serve as electrical signatures to various faulty operations and facilitate in devising a monitoring algorithm for the system. A model-based approach has been adopted to represent a P.V. system, using a one-diode model of a practical P.V. cell, developed in MATLAB/ Simulink. The modelled system comprises two arrays, while each array has two panels in series. It was simulated for various operating conditions: healthy condition represented by STC (Standard Testing Condition), O.C. (open-circuited), soiling, P.S. (partial-shading), H. S. (panels hotspots) and P.D. (panels degradation) conditions. For the analysis of I-V curves under these conditions, six derived parameters were selected: Vte (equivalent thermal voltage), MCPF (maximum current point factor), Ri (currents ratio), S (slope), and Dv and Di (voltages and currents differences, respectively). Using these parameters, data of the actual system under various conditions were compared with its model-generated data for healthy operating conditions. Thresholds were set for each parameter's value to mark normal operation range. It was observed that almost each considered fault creates a unique combination of sensitive parameters whose values exceeds the pre-defined thresholds, creating an electrical signature that will appear only when the corresponding conditions on the system are achieved. Based on these signatures, an algorithm has been proposed in this study which aims to identify and classify the considered faults. In comparison to other such studies, this work has been focused on those sensitive parameters for faults identification which shows

**Funding:** The authors received no specific funding for this work.

**Competing interests:** The authors have declared that no competing interests exist.

greater sensitivity and contribute more to creation of unique sets of sensitive parameters for considered faults.

## 1. Introduction

Energy—being the necessity for economic development—has played an essential role in making up current-day civilization. The living standards of a country can be judged from energy consumption per person living there [1]. Owing to technological advancements and growing population, need for energy is increasing which is provided through multiple energy sources broadly classified into non-renewable and renewable resources. Due to the depleting nature of the former and its negative environmental impact, the world is shifting towards the latter and its related technologies [2]. Renewable energy sources are intermittent sources, hence unreliable [3]. Efforts are on their way to increase its efficiency and reliability [4]. Among renewables, the most significant and widely used energy is solar energy- available in heat and light from the sun [5]. It can be harnessed using a concentrated solar power plant, utilizing solar heat energy [6]. While, solar energy in the form of light can be harnessed through P.V. modules/panels that work on converting sunlight (light-photons energy) into electrical energy [7]. The P.V. market is growing exponentially due to its fascinating features such as cutback in the production charges, net metering, off-grid installation in remote areas, the long-life span of the P.V. modules, and supporting policies [5, 8–10]. These enthralling features of the P.V. market have made the return on investment more interesting.

Nevertheless, like any other process, a P.V. system operation is vulnerable to certain limitations and faults [3]. These limitations include module's energy conversion efficiency and inverter efficiency, both are the salient components of a P.V. system [11, 12]. Besides these limitations, a P.V. system is vulnerable to certain faults during its operation, thus restricting it to operate below optimal power levels and may also lead to equipment and personnel damage if remained undetected [13]. A PV system, once installed, is expected to operate without human intervention. P.V. modules operate in an external environment, hence exposed to environmental conditions such as sunlight, heat, dust, humidity, shadowing etc. Although solar irradiance directly relates to P.V. output, yet some wavelengths have a negative impact on its operation, such as ultraviolet and infrared are instead added as heat [14]. The accumulation of dust and dirt on the modules, known as soiling, compromise the output efficiency of a P.V. system [15]. The effect of shadowing also has a negative impact on its output [16]. Besides direct environmental impacts, a P.V. system is vulnerable to open-circuit faults, short-circuit faults, mismatch faults and partial degradation of modules [17]. More often these faults remain undetected, therefore, it is required to monitor the system to identify such faults and clear them if needed. Monitoring techniques to perform this task are broadly classified into two classes, i.e. thermal/visual and electrical methods [18]. Former requires frequent visits, is time consuming, manual, and least efficient. While latter works on electrical quantities known as electrical signatures derived from system's output.

The methodology opted in this work, to monitor a P.V. system against various contingent situations, belongs to electrical methods. It relies on the analysis of the I-V characteristics of the P.V. plant, using a model-based approach. By comparing these characteristics of an actual system with its modelled system with help of derived parameters from it, system's health could be inferenced. Such an approach has been opted in this work using One-diode model with improved parameters for modelling a P.V. system, as it is uncomplicated and accurate enough to represent a practical P.V. plant operation [19]. The model has been implemented and

analyzed in MATLAB/Simulink. For creating electrical signatures, resulting from comparison of modelled system and actual system characteristics, six derived parameters from the I-V characteristics of a P.V. system were selected among others due to their better response. Variance and Sensitivity of these parameters were analyzed for different faulty conditions such as soiling, P.S., O.C., H.S. and P.D. Results and observations showed that these parameters better act in creating electrical signatures to these faults and justify their role in devising a monitoring algorithm. Based on them, an algorithm was derived that assigns electrical signatures to each faulty condition and successfully classifies all the considered faults.

## 2. Modelling

A One-diode model of a practical P.V. cell is given in Fig 1. The shunt (R*sh*) and series (R*s*) resistances account for the losses in a practical P.V. cell operation. The equation that describes the One-diode model is given by Eq (1) in which Iph is photon current generated by incident light, *Id* is the shunt diode current in the model, *Ish* is the shunt resistance path/leakage current, and *I* and V are the terminal current and voltage at a P.V. cell, respectively.

$$I = Iph - Id - Ish \tag{1}$$

$$I = Iph - Io \left[exp(q(V + I \, Rs))/A \, K_B \, T) - 1\right] - (V + IRs)/Rsh \tag{2}$$

While observing Eq (2), *Io* is the diode saturation current, T is panel temperature in kelvin. Where, q ($1.602 \times 10^{-19}$ C) is charge on an electron, *K.B.* ($1.38065 \times 10^{-23}$ J/K) is Boltzmann constant and *A* is the diode ideality factor of a P.V. cell. For several cells connected in series (Ns), to make a panel/module, the equation becomes,

$$I = Iph - Io \left[exp(V + I \, Rs)/A \, Vt) - 1\right] - (V + I \, Rs)/Rsh \tag{3}$$

Where as

$$Vt = \frac{Ns \times K_B \times T}{q} \tag{4}$$

Diode saturation current, *Io*, is given by Eq (5).

$$Io = \frac{Iscn + ki\Delta T}{exp \left(\frac{Vocn + kv\Delta T}{AVt}\right) - 1} \tag{5}$$

*kv* and *ki* are temperature coefficients for voltage and current, respectively, provided by the manufacturer datasheet, along with open-circuit voltage (*Vocn*) and short-circuit current

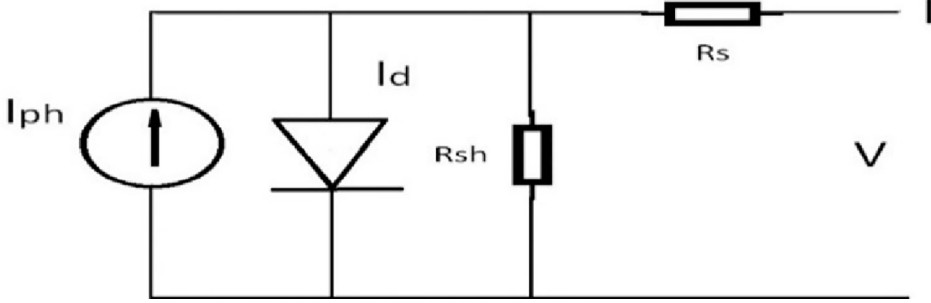

**Fig 1. One-diode model of a practical P.V. cell.**

(*Iscn*) at STC. Eq (5) replaces previous equations for diode saturation current and represents Sensitivity of voltage and current to temperature changes more prominently, thus improving the model response [19]. Photon generated current of a panel is given by Eq (6), while Eq (7) represents photon induced current at STC.

$$Iph = [Iphn + ki(T - Tn)]\frac{G}{Gn} \tag{6}$$

$$Iphn = \left({}^R s + Rsh/_R sh\right) \times Iscn \tag{7}$$

In Eq (6), G represents available solar irradiance, Gn represents solar irradiance at STC (i.e., 1000w/m$^2$), and Tn (in Kelvin) represents the temperature of the panel at STC. These equations can be solved using data from the P.V. panels manufacturer datasheet along with irradiance and temperature values for any operating condition. Literature study reveals that a pyranometer at the panel's surface can estimate the irradiance very well, while temperature of the panels can be estimated using the empirical formula in [20], thus reducing the number of sensors. Based on these equations of the One-diode model, a P.V. system consisting of two arrays, each having two panels in series (i.e., 2×2 system), was developed in Simulink and simulated for various conditions to obtain their corresponding I-V curves [21]. Furthermore, for the analysis of I-V curves obtained, MATLAB has been used where three crucial points on the I-V curves i.e., short-circuit point, open-circuit point and MPP (maximum power point) were calculated. From these three points of the I-V curve, their derived parameters were calculated for each condition.

## 3. Methodology

The methodology opted in this work, for monitoring a P.V. system, has been based on comparison of I-V characteristics of an actual system with that of its modelled system for the same inputs (i.e., G and T) available to the actual system. For any inputs of G and T, the modelled system represents healthy condition response of a system, while the actual system represents an actual operating condition that could be healthy or faulty. By comparing the I-V curves of both systems, the health of the actual system can be diagnosed. To compare I-V curves of both systems, only few points on the curve are required that can better represent the overall behavior of the curve as well as show prominent and effective change with changing conditions. This work has considered three such vital points on the I-V curve (i.e., short-circuit point, open-circuit point and maximum power point). Now, to find relative changes between these points, some parameters were derived from them and their responses were observed and compared. After a rigorous analysis of different derived parameters response towards various faults, six parameters, given by Eqs (7–12), were selected as they showed prominent sensitivity to various changing operating conditions and their combination of sensitive parameters to various faults allowed to assign different electrical signatures to them. Based on the analysis of the parameters' response towards different operating conditions, a range of values close enough to healthy value for a parameter was defined as the threshold region. The threshold region represents a normal/inactive region, while outside it represents a sensitive/active region for a parameter. It represents a range of values where a parameter is considered least sensitive and is unable to mark a difference between various faults as each fault have some effect on almost each parameter and causes it to deviate from healthy value. Therefore, to mark difference between various faults, only those values will be considered responsible for sensitivity which represents large deviation and lies outside the threshold region. This way, not every fault will make every parameter sensitive but only those will become sensitive who had values outside threshold,

thus leading to distinguishing various faults. For each parameter such thresholds were identified that helped in differentiating various conditions. Thus, a parameter will be considered sensitive to a fault if it exceeds its corresponding threshold in response to it. A set of sensitive parameters comprises an electrical signature for a fault. The six derived parameters considered in this work for creating and assigning distinguishable electrical signatures to faulty conditions are defined below.

The first parameter considered here is Vte (equivalent thermal voltage). It is sensitive to faults of heterogenous impact (affecting MPP) as well of homogenous impact (affecting short-circuit point). Also, the secondary diagnostic role of this parameter is in the approximation of diode ideality factor deterioration factor [22]. Vte can be derived from the I-V characteristics using Eq (8), where Isc and Voc stand for current at short-circuit point and voltage at open circuit point, respectively, while Imp and Vmp represent current and voltage at the maximum power point.

$$Vte = \frac{(2Vmp - Voc)(Isc - Imp)}{Imp - (Isc - Imp)ln\left(\frac{Isc-Imp}{Isc}\right)} \tag{8}$$

The second parameter, considered here, is MCPF (maximum current point factor). MCPF is usually sensitive to faults of homogenous impact such as soiling. It can be derived using Eq (9), which includes solar irradiance as an input along with current at short circuit point.

$$MCPF = \frac{G}{Gn \times Isc} \tag{9}$$

The third derived parameter from the I-V curve is the slope (S) between short-circuit point and the open-circuit point. It represents the relative movement of these two points and will show Sensitivity to faults that impact any of these two points. It is given by Eq (10).

$$S = \left(-\frac{Isc}{Voc}\right) \tag{10}$$

The remaining three parameters were derived from I-V characteristics using simple arithmetic. They were obtained by taking ratio(s) [23], represented by Eq (11), and difference(s) between the three important points on the I-V curve represented by Eqs (12) and (13).

$$Current\ Ratio\ (Ri) = \frac{Isc}{Imp} \tag{11}$$

$$Voltage\ Difference\ (Dv) = Voc - Vmp \tag{12}$$

$$Current\ Difference\ (Di) = Isc - Imp \tag{13}$$

As stated earlier, these parameters were calculated for various operating conditions such as healthy condition (at STC), soiling, P.S. (partial shadings), H.S. (panels hotspots), P.D. (panels degradation) and an O.C. (open-circuit condition). According to the panels configuration of the system (2×2), some of these faults, usually of heterogeneous impact, can appear on the system in multiple distinct ways. This paper has considered some of them and classified them as distinct faults. Partial shading faults were classified as P.S. (1) (partial shading on one panel only), P.S. (2) (on single array only) and P.S. (3) (on single panel in each array). At the same time, P.S. (3) was further classified as P.S. (3.1) and P.S. (3.2) based on the intensity of shading, as P.S. (3) shows an abrupt change in response to an increase in shading intensity beyond a certain value of available G to the shaded panels. P.S (3.1) represents partial shading on single

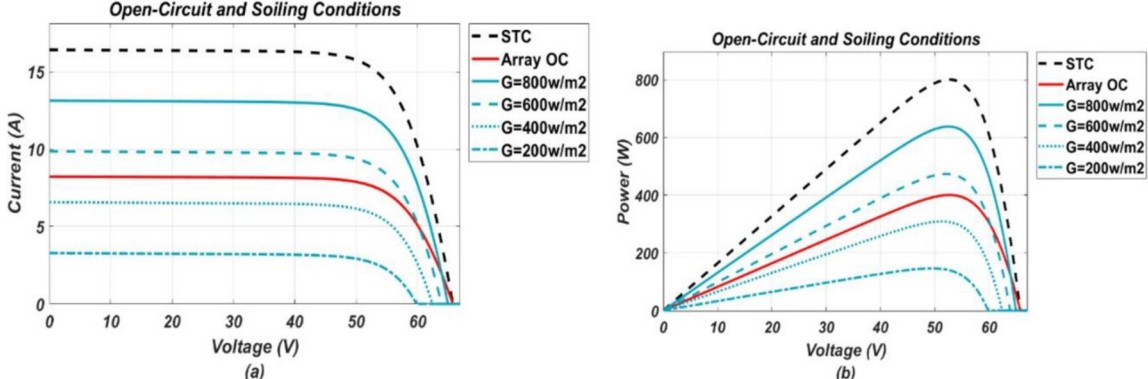

**Fig 2.** (a) and (b), I-V and P-V graphs respectively, representing response of P.V. system under open-circuited and soiling conditions of different intensity levels.

panel in each array when the fault intensity on shaded panels is such that the available G is not less than almost 600w/m$^2$ and the same fault is represented by P.S (3.2) for values of G below 600w/m$^2$. Likewise, hotspot faults were classified as H.S. (1) (all panels' hotspots), H.S. (2) (single array hotspots), H.S. (3) (single panel hotspot in each array), and H.S. (4) (single panel hotspot in the whole system). Similarly, for panels degradations, this sequence (1–4) represents all panels, single panel, single array, and single panel in each array degraded, respectively. Thus, a total of 15 conditions were simulated on the modelled system, one of which is a healthy condition. All of 14 faulty conditions were simulated starting from low intensity of impact to high-intensity impact. Based on the results analysis and setting proper thresholds, sensitive parameters for various conditions were identified to create corresponding electrical signatures and devise a monitoring algorithm.

## 4. Results and discussion

The model-based system, developed in MATLAB/Simulink, was simulated for various operating conditions stated earlier. Response of the P.V. system under these various considered conditions were obtained in the form of I-V and P-V (power-voltage) graphs, respectively. From the P-V response, it was found that the system tends to operate below optimal output power level when under fault and output power drops more as fault intensity increases. While I-V

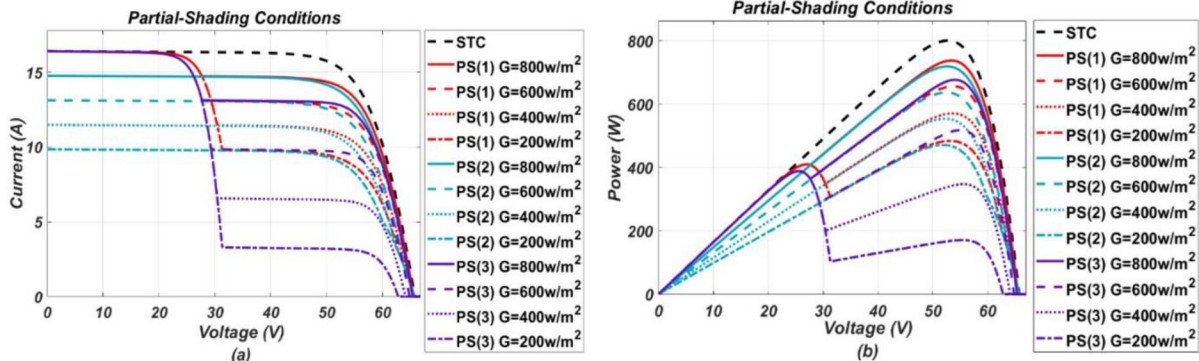

**Fig 3.** (a) and (b), I-V and P-V graphs respectively, representing response of P.V. system under partial shading conditions of different intensity levels.

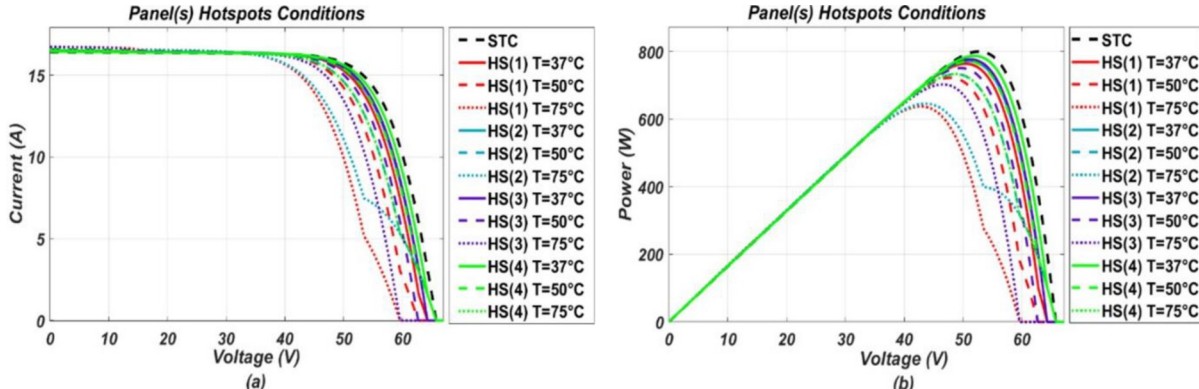

**Fig 4.** (a) and (b), I-V and P-V graphs respectively, representing response of the P.V. system under panel(s) hotspots conditions of different intensity levels.

response shows, different faults have different impact on I-V characteristic points. These responses, in the form of graphs, are presented below in the Figs 2A–5B.

It can be seen from the I-V curves that soiling, O.C. and P.S. (2) affect the short circuit current by reducing it and along with it MPP also gets reduced from its referenced healthy position. Soiling is a kind of homogenous fault as it is expected to spread uniformly on all panels of the system. Similarly, P.S. (2) and O.C. can also be classified as homogenous nature faults as they affect a specific portion of the system uniformly, and their resulting impact resembles that of soiling. On contrary, partial shadings such as P.S. (1) and P.S. (3) are considered heterogeneous faults as they affect the system non-uniformly. Partial shadings of such type do not reduce the short circuit current, rather only affect MPP by reducing current at MPP. As a result, two steps are formed in the I-V curve as shown in the Fig 3A in which the upper step resembles a healthy operation while the lower step represents a faulty operation. For H.S. conditions, they have a very slight impact on the short circuit current by increasing it a little bit while showing more impact on the open-circuit voltage by decreasing it from the referenced healthy condition value. Furthermore, for P.D. conditions, they tend to slightly reduce the MPP only by changing slope at the knee point on the I-V curve, while the other two points remain unchanged.

Now, responses of the six derived parameters to these various faulty conditions are presented in the graphs provided in the figures below. These graphs were obtained by plotting the response of each parameter under various faulty conditions against increasing fault intensity levels.

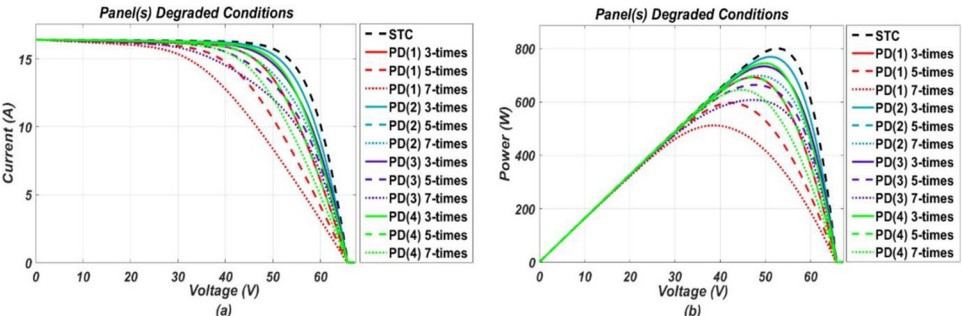

**Fig 5.** (a) and (b), I-V and P-V graphs respectively, representing response of the P.V. system under panel(s) degradation conditions of different intensity levels.

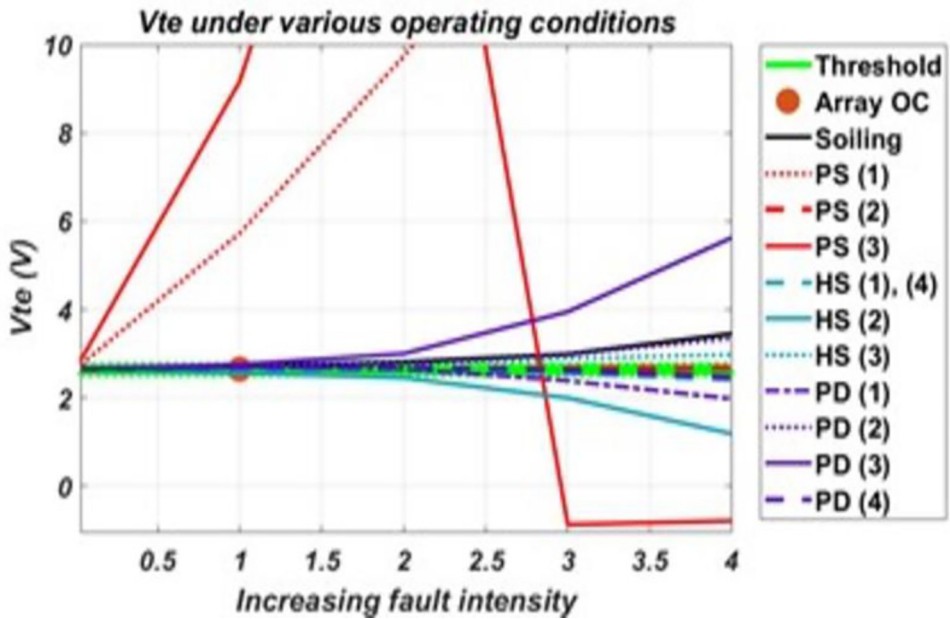

**Fig 6. The response of Vte to various conditions and their varying intensity levels.**

It can be seen from the graph in Fig 6, Vte is sensitive to faults both of homogenous impact (usually affecting short-circuit point) as well of heterogenous impact (usually involving MPP). Apart from four conditions represented by the lines in green region, which represent the threshold region, Vte shows Sensitivity to all other considered faults. Threshold region was identified from the analysis of the parameter's values for faulty conditions, outside which a fault intensity will be considered pretty enough to make the parameter sensitive. It was obtained by subtracting healthy value for the parameter from the lowest value for which the parameter is considered sensitive on either side of the healthy value. From the data analysis, Vte shows significant variance even at low fault intensities to conditions such as P.S. (1) and P.S. (3.1), which are of heterogeneous impact. Furthermore, analysis shows that four different regions for values outside the pre-defined threshold can be set: values very much greater than threshold (Vte > 3.5 for this system) as in case of P.S (1) and P.S (3.1), values that are slightly greater than threshold (3.5 > Vte > threshold) such as for faults H.S (3), soiling, PD (2) and PD (3), values less than the threshold but not negative (0< Vte <threshold) as incase of HS (2) and PD (1), and negative values such as for P.S (3.2). Setting these multiple thresholds for Vte helped in distinguishing various operating conditions.

Response of MCPF, in Fig 7, shows that it is sensitive to faults of homogenous impact such as soiling, P.S. (2) and array O.C. It shows more significant variance and increasing trend in values from the threshold range to these conditions. Also, it was found that this parameter shows a small variance and decreasing trend in values to all H.S. conditions. Hence, two regions for values outside the threshold range can be set, i.e., one above/greater than the threshold and the other below/less than it.

It can be seen from Fig 8 that the slope parameter shows sensitivity to faults of homogenous impact such as soiling, array O.C and P.S. (2). These faults tend to reduce short-circuit current, causing the parameter to adopt decreasing trend from the pre-defined threshold as the fault intensity increases. Also, hotspot conditions such as H.S. (1) and H.S. (3) tends to increase the short-circuit current and decrease the open-circuit voltage slightly, makes the parameter

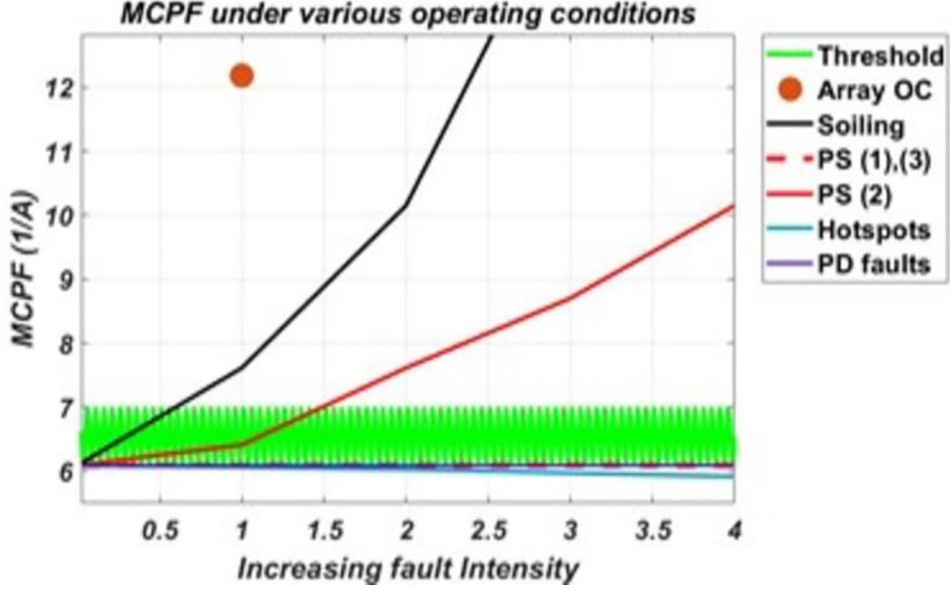

**Fig 7. The response of MCPF to various conditions and their varying intensity levels.**

sensitive by adopting an increasing trend. Hence, two regions for values outside the threshold have been selected.

Results, in Fig 9, show that Ri is sensitive to faults of heterogenous impact such as P.S. (1) and P.S. (3), more specifically partial shading faults of such nature. These faults affect MPP by reducing current at MPP while keeping the short circuit current almost constant, resulting in a larger value of Ri. Hence, Ri shows sensitivity to these faults only.

It can be seen from the graphs for Dv and Di in the Figs 10 and 11, respectively, like Vte; these two derived parameters also show Sensitivity to faults of both homogenous and

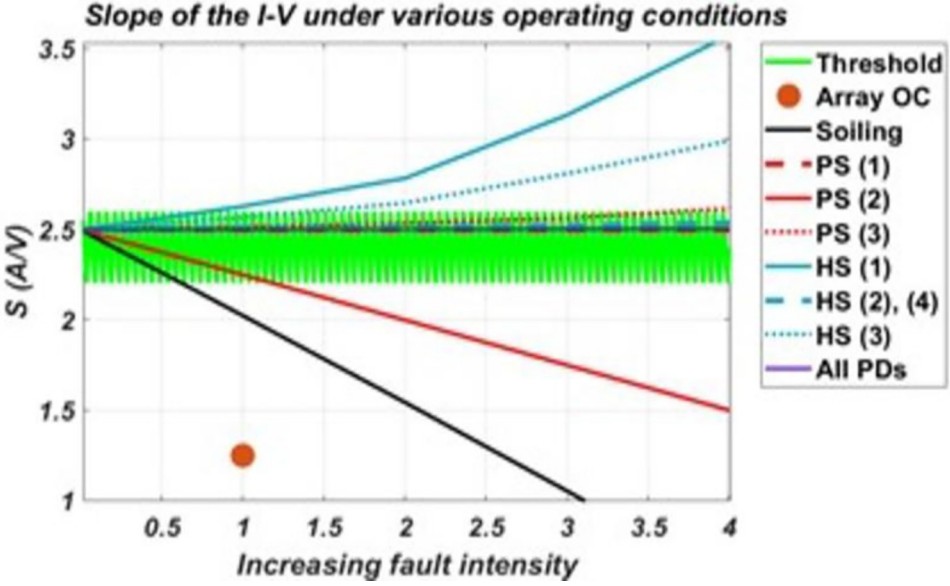

**Fig 8. Response of S (slope) to various conditions and their varied intensity levels.**

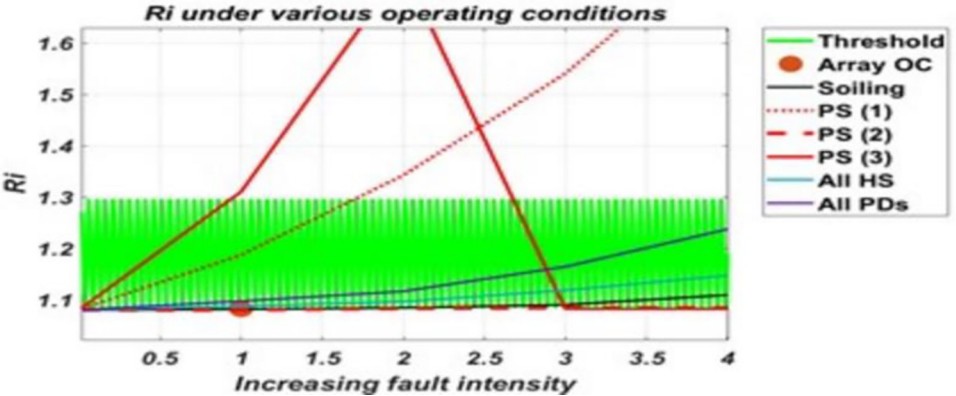

**Fig 9. The response of Ri to various conditions and their varying intensity levels.**

heterogeneous nature among the considered faulty conditions. Fig 10 shows that Dv is not sensitive to O.C., P.S. (2), H.S. (1) and H.S. (3). Dv shows Sensitivity to all other considered conditions other than these faults, and two regions outside the threshold can be set for Sensitivity, i.e., regions above/greater and below/less than the threshold. Similarly, from the Fig 11, it can be seen that Di shows Sensitivity to all considered conditions except P.D. (2), P.D. (4) and all hotspots' conditions. For H.S. (1) and H.S. (2), Di will become Sensitivity only when the fault intensity increases very much, as seen from the graph. But before reaching such intensity, the system is able to detect them much earlier from other corresponding sensitive parameters; that's why Di has not been considered sensitive to them.

As shown by these graphs for parameters, P.S. (3) shows an abrupt change when the fault intensity increases on the faulty portion and received irradiance to the faulty portion becomes almost half of that of a healthy portion. For this reason, it had been further classified as P.S.

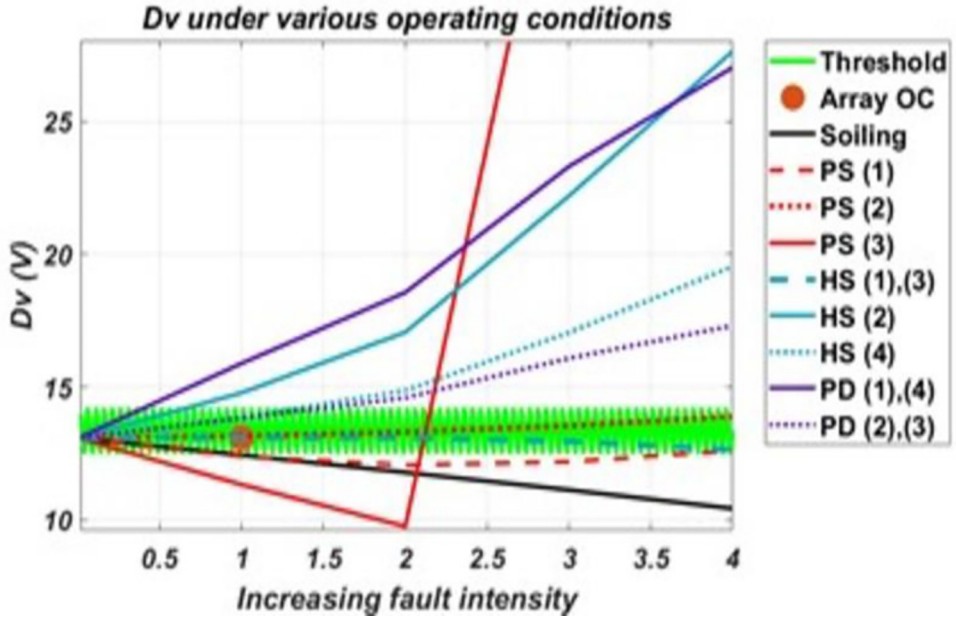

**Fig 10. Response of Dv to various conditions and their varied intensity levels.**

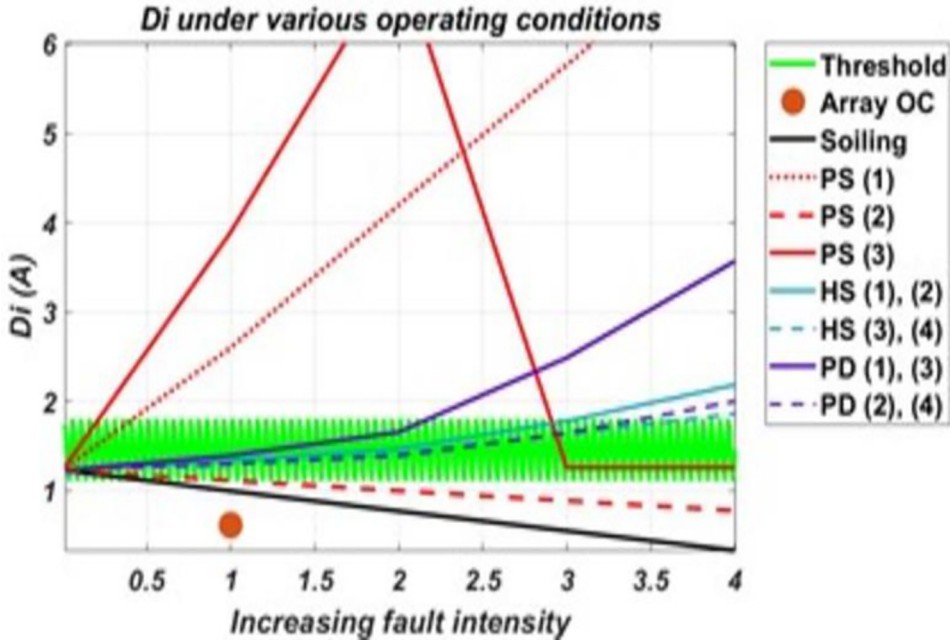

**Fig 11. The response of Di to various conditions and their varying intensity levels.**

(3.1) and P.S. (3.2), as the parameters' Sensitivity are changed for both. Also, it can be seen from these graphs that some parameters which although acts as sensitive parameters towards a particular fault, will remain insensitive at low-intensity levels. Such soiling will be detected when its intensity is such that the difference between available and received irradiance exceeds $230w/m^2$. Similarly, P.D. will be detected when cells' series and shunt resistances degradation approach 4–5 times that of a non-degraded cell/panel. For hotspots, they will be detected when faulty panels' temperature approaches near twice that of a healthy panel(s) in the system. While in partial shading conditions, all except P.S. (2) are detected even at low-intensity fault levels. P.S. (2) will be detected at intensity levels same as in the case of soiling. From the analysis of these results, it was found that using these derived parameters, the various considered conditions on the system can be classified as almost each condition has a different combination of sensitive parameters. Thus, creating different sets of sensitive parameters for the considered conditions and allows assigning electrical signatures to these them. Only two cases were found where the same set of parameters following the same trend represents two faulty conditions (i.e., for P.S. (1), P.S. (3.1) and for O.C., P.S. (2)). Using these electrical signatures, an algorithm has been proposed in this paper for monitoring the system against the considered faulty conditions, given in Fig 12. This algorithm successfully classifies and identifies the faulty conditions and monitors the system against the considered faulty conditions.

The method presented here at this work has, first of all, the advantage of being an electrical method of faults identification using output values of the system, hence, leads to automatic faults monitoring system. Secondly, this work has been focused on improving faults identification from other electrical methods by selecting those parameters whose responses can very well contribute to create unique sets of sensitive parameters for various faults and also by introducing few parameters such as Di and Dv that can contribute well in the desired task. In [24], the three points on I-V curve, already discussed here, had been employed for performing the desired task of monitoring along with slopes between these points, while this paper had addressed the parameters derived from these points. Hence, the derived parameters have the advantage of having diversity,

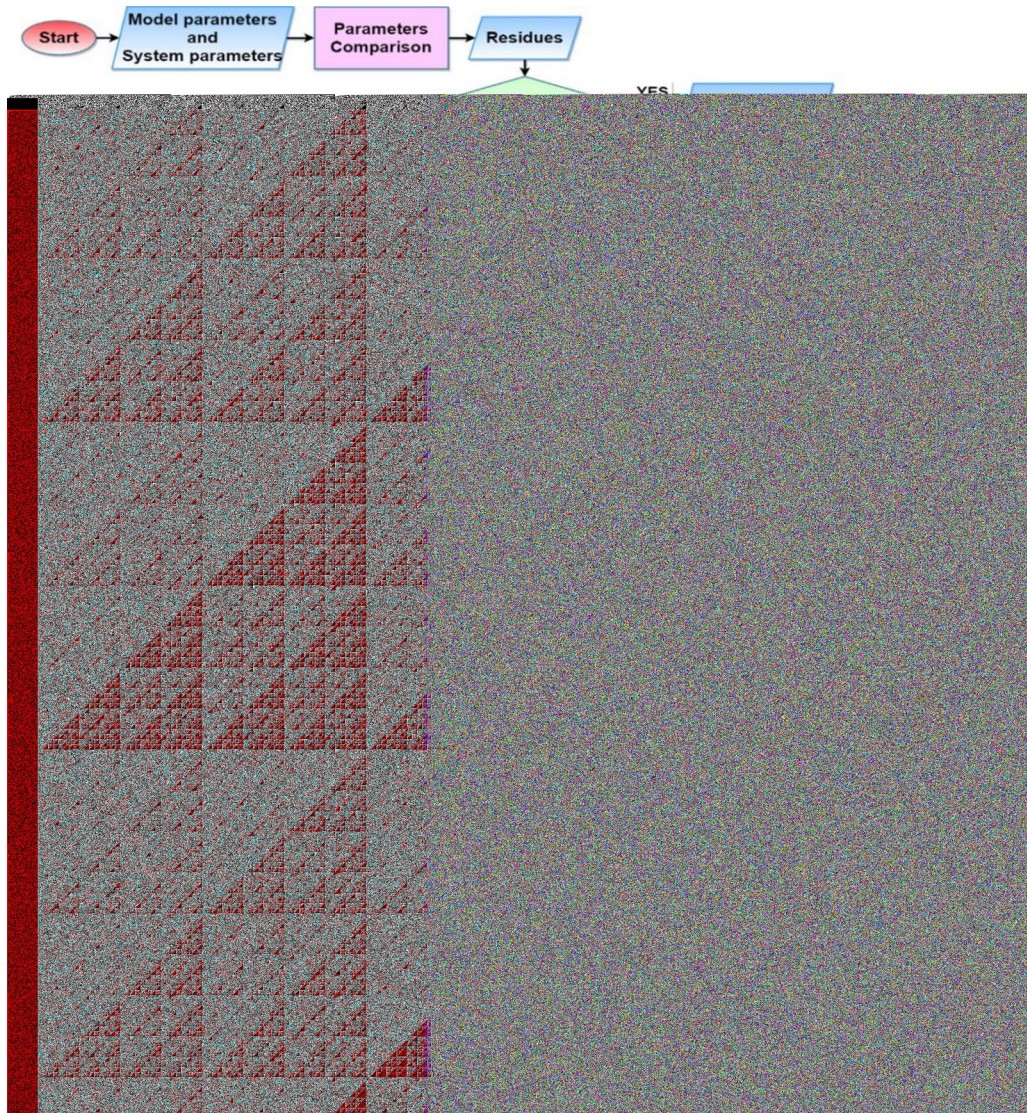

**Fig 12. A diagnostic algorithm based on the electrical signatures obtained from the six derived parameters for monitoring the P.V. system.**

leading to creating many numbers of such derived parameters making it easy to distinguish various faults. Likewise, in [23] some derived parameters were considered for the desired task, while this work took into account a total of 14 such parameters, including those considered in [23], at the beginning. Then, based on the sensitivity and contribution to create unique electrical signatures, only 6 derived parameters were finally chosen which showed promising results for performing the task of distinguishing and identifying various faults. These parameters were selected based on assigning unique electrical signatures to various considered faults and was found they are most suitable than the ones considered in previous work.

## 5. Conclusion

For monitoring a P.V. system, the approach used here has been based on comparing the response of an actual system with that of its modelled system. For modelling a P.V. system, a

One-diode model with improved parameters has been used, which better represented the typical response of a P.V. system. The model-based approach proved an efficient and effective way of studying and analyzing P.V. system response under various operating conditions. For the comparison of response of both systems, six parameters were derived from the I-V characteristic points. These parameters were chosen based on their sensitivity to changing operating conditions and ability to make sets of sensitive parameters that could assign signatures to various faults. Almost 14 kinds of faulty scenarios along with healthy conditions were observed in this paper and were found that these six parameters successfully classify them and allow the assignment of different signatures to them. An algorithm has been proposed using these signatures with the aims to detect a fault and distinguish it among the considered faults, otherwise state undefined condition/fault. This successfully shows the efficacy of electrical signatures in monitoring a P.V. system, comprising an electrical method of monitoring a P.V. system.

Future work is recommended to extend the scope of this approach to big P.V. systems (greater than 2×2) and more numbers of faulty conditions (such as arc faults, ground faults, etc.). For a big system and more faults, the total possible faulty conditions will likely increase, and so will the corresponding electrical signatures. It is also recommended to include some new parameters—derived from I-V points—that could be more sensitive to faults, especially to H.S. and P.D. conditions. For P.D. conditions, the relation between sensitivity of sensitive parameters and percent degradation should be calculated, and factors other than degradation of series and shunt resistances of cell/panel should also be considered. Also, this work has not considered the appearance of two faults simultaneously. Although the algorithm presented here will detect the appearance of simultaneous faults but cannot classify/identify them. Furthermore, it is also recommended to calculate the threshold range in percentage i.e., to calculate the deviation from a nominal value in percentage for each parameter. In addition to these limitations, this work has also not addressed the localization of faults.

## Supporting information

**S1 Data.**
(MAT)

**S1 File.**
(DOCX)

## Acknowledgments

I acknowledge that author Muhammad Rizwan Siddiqui is removed from the author list, while Abdullah Mohammed is included. All the updated authors in the list have contributed to the study and do not violate the journal authorship criteria.

## Author Contributions

**Conceptualization:** Muhammad Adnan Khan, Zubair Ahmad Khan, Abdullah Mohammed.

**Data curation:** Shahbaz Khan.

**Formal analysis:** Muhammad Adnan Khan, Khalid Khan, Shahbaz Khan, Abdullah Mohammed.

**Funding acquisition:** Zubair Ahmad Khan, Shahbaz Khan, Abdullah Mohammed.

**Investigation:** Muhammad Adnan Khan, Zubair Ahmad Khan, Abdullah Mohammed.

**Methodology:** Muhammad Adnan Khan, Adnan Daud Khan, Shahbaz Khan.

**Project administration:** Khalid Khan, Adnan Daud Khan.

**Software:** Muhammad Adnan Khan.

**Supervision:** Adnan Daud Khan, Zubair Ahmad Khan, Abdullah Mohammed.

**Validation:** Adnan Daud Khan.

**Writing – review & editing:** Khalid Khan, Abdullah Mohammed.

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
