## [Decision Letter · Decision Letter 0]

1 Sep 2021

PONE-D-21-20995

A model-based approach for detecting and identifying faults on DC side of a PV system using electrical signatures from I-V characteristics

PLOS ONE

Dear Dr. Khan,

Thank you for submitting your manuscript to PLOS ONE. After careful consideration, we feel that it has merit but does not fully meet PLOS ONE’s publication criteria as it currently stands. Therefore, we invite you to submit a revised version of the manuscript that addresses the points raised during the review process.

We look forward to receiving your revised manuscript.

Kind regards,

Lei Chen, Ph.D.

Academic Editor

PLOS ONE

Journal Requirements:

Additional Editor Comments:

Please carefully address the comments of two reviewers to improve your paper.

Reviewers' comments:

Reviewer's Responses to Questions

**Comments to the Author**

1. Is the manuscript technically sound, and do the data support the conclusions?

Reviewer #1: Partly

Reviewer #2: Yes

2. Has the statistical analysis been performed appropriately and rigorously? 

Reviewer #1: I Don't Know

Reviewer #2: Yes

3. Have the authors made all data underlying the findings in their manuscript fully available?

Reviewer #1: Yes

Reviewer #2: Yes

4. Is the manuscript presented in an intelligible fashion and written in standard English?

Reviewer #1: Yes

Reviewer #2: Yes

5. Review Comments to the Author

Reviewer #1: 1. There are places in the article that do not indicate which equation to cite, such as the paragraph above Equation 7.

2. What does P.S. (3.1) mean in the descriptive paragraph of Figure 6?

3. In the analysis of Figure 6, it is mentioned that four different regions can be set for values beyond the predefined thresholds, but it is not clearly described what the four regions are and how to define the boundary between the four regions.

4. Appropriately add some relevant descriptions on how to determine the threshold range.

5. Add some comparisons with other scholars' work on electrical methods.

6. Further elaborate on the specific advantages or innovations of this method.

Reviewer #2: 1. The introduction of this submission should be improved further. The key differences between your work and previous studies should be clarified. A point-to-point way is recommended to state the main contributions.

2. PV modeling section is very traditional, and this reviewer suggests this section can be shorten properly.

3. The writing style of this work likes a technical report, but not a scientific paper.

4. A comparison of your method and other work should be carried out.

5. Most of the figures have poor quality, such as, lacking of unit.

6. PLOS authors have the option to publish the peer review history of their article (what does this mean?). If published, this will include your full peer review and any attached files.

Reviewer #1: No

Reviewer #2: No

---

## [Author Response · Author response to Decision Letter 0]

26 Sep 2021

Editor’s comments,

Comments to the Author

Thank you for submitting your manuscript to PLOS ONE. After careful consideration, we feel that it has merit but does not fully meet PLOS ONE’s publication criteria as it currently stands. Therefore, we invite you to submit a revised version of the manuscript that addresses the points raised during the review process. We look forward to receiving your revised manuscript.

Author’s response: The corresponding author, on behalf of all the co-authors, would like to thank the editor for the positive feedback and encouragement. The revised version of the manuscript has been updated according to the guidelines provided by the PLOS ONE style requirements. 

Author’s response: The funders had no role in study design, data collection and analysis, decision to publish, or preparation of the manuscript.

Author’s response: No funding or any financial grant has been received for conducting this study. 

b) State what role the funders took in the study. If the funders had no role in your study, please state:

Author’s response: The funders had no role in study design, data collection and analysis, decision to publish, or preparation of the manuscript

Author’s response: The authors received no specific funding for this work. 

Author’s response: There are no ethical restrictions on sharing a de-identified data set. 

Author’s response: The data set is uploaded in the review as a Supporting Information file. 

4. Please amend your list of authors on the manuscript to ensure that each author is linked to an affiliation.

Author’s response: The list of authors is provided with affiliation in the given order, which is requested to be considered in the revised version of the manuscript. 

i. Muhammad Adnan Khan1, 

 Affiliation: Research Associate, Center for Advanced Studies in Energy University of Engineering and Technology, Phase 5, opposite to Sui Northern Gas Pipeline office, Postal address 25000, Peshawar. 

ii. Khalid Khan2* 

Affiliation: Research Associate, Center for Advanced Studies in Energy University of Engineering and Technology, Phase 5, opposite to Sui Northern Gas Pipeline office, Postal address 25000, Peshawar.

iii. Adnan Daud Khan3*

Affiliation: Dean Faculty Renewable Energy, Center for Advanced Studies in Energy University of Engineering and Technology, Phase 5, opposite to Sui Northern Gas Pipeline office, Postal address 25000, Peshawar.

iv. Zubair Ahmad Khan4

Affiliation: Professor at Department of Mechatronics, University of Engineering and Technology, Phase 5, opposite to Sui Northern Gas Pipeline office, Postal address 25000, Peshawar.

v. Shahbaz Khan5

Affiliation: Lab Engineer at Department of Mechatronics, University of Engineering and Technology, Phase 5, opposite to Sui Northern Gas Pipeline office, Postal address 25000, Peshawar.

vi. Muhammad Rizwan Siddiqui6

Lecturer at Capital University of Science and technology (CUST), Islamabad Expressway, Kahuta، Road Zone-V Sihala, Islamabad, Islamabad Capital Territory.

5. PLOS authors have the option to publish the peer review history of their article. If published, this will include your full peer review and any attached files.

Author’s response: The authors agree to publish the review history of the article.

Please carefully address the comments of two reviewers to improve your paper.

Editor’s comments,

Concern #1:

Is the manuscript technically sound, and do the data support the conclusions?

Reviewer #1: Partly

Author response: The authors appreciate the positive remarks of the reviewer #1 and would like to reflect on any particular section of the manuscript to improve its quality, if highlighted by the reviewer. 

Reviewer #2: Yes

Author response: The authors appreciate the positive response of the reviewer #2.

Concern #2:

Has the statistical analysis been performed appropriately and rigorously?

Reviewer #1: I Don't Know

Author response: The authors would be happy to elaborate further on the techniques used in the article, provided specific questions are asked to be addressed. 

Reviewer #2: Yes

Author response: The authors appreciate the feedback of the reviewer #2.

Concern #3:

Have the authors made all data underlying the findings in their manuscript fully available?

The PLOS Data policy requires authors to make all data underlying the findings described in their manuscript fully available without restriction, with rare exception (please refer to the Data Availability Statement in the manuscript PDF file). The data should be provided as part of the manuscript or its supporting information or deposited to a public repository. For example, in addition to summary statistics, the data points behind means, medians and variance measures should be available. If there are restrictions on publicly sharing data—e.g., participant privacy or use of data from a third party—those must be specified.

Reviewer #1: Yes

Author response: The authors appreciate the positive feedback of the reviewer #1.

Reviewer #2: Yes

Author response: The authors appreciate the positive feedback of the reviewer #2.

Concern #4:

Is the manuscript presented in an intelligible fashion and written in standard English?

Reviewer #1: Yes

Author response: The authors appreciate the positive feedback of the reviewer #1.

Reviewer #2: Yes

Author response: The authors appreciate the positive response of the reviewer #2.

5. Review Comments to the Author

Please use the space provided to explain your answers to the questions above. You may also include additional comments for the author, including concerns about dual publication, research ethics, or publication ethics. 

Reviewer #1: 

1. There are places in the article that do not indicate which equation to cite, such as the paragraph above Equation 7.

Author’s response: The author appreciates reviewer #1 comment on this issue, as it was mistakenly left by author in the earlier draft. This mistake has been rectified and all equations are numbered properly as well as cited properly in the new draft. Equations after correction are highlighted yellow in the new draft.

2. What does P.S. (3.1) mean in the descriptive paragraph of Figure 6?

Author’s response: This question has been answered more properly and the issue has been addressed in the new draft in the last paragraph of methodology section, being highlighted in yellow there. Hope this will answer reviewer’s question.

3. In the analysis of Figure 6, it is mentioned that four different regions can be set for values beyond the predefined thresholds, but it is not clearly described what the four regions are and how to define the boundary between the four regions.

Author’s response: To answer this question, a brief description has been added to the new draft just below fig. (6), in highlighted text. This description has elaborated, in a brief way, on threshold levels and how they were selected. Furthermore, all the four regions regarding “Vte” are described and the boundaries between them are clearly defined in that. However, the focus of this paper is on checking the sensitivity of parameters that could serve in creating electrical signatures. Hence, it was found that Vte does the job by responding to various operating conditions in ways different enough from each other to differentiate between faults, which description tells clearly.

4. Appropriately add some relevant descriptions on how to determine the threshold range.

Author’s response: This question has already been answered in question 3. Apart from that, threshold levels are defined more clearly in first paragraph of Methodology section, text highlighted in yellow. Hope that will answer the question.

5. Add some comparisons with other scholars' work on electrical methods.

Author’s response: A comparison with other scholars’ work has been made at the end of results and discussions section. Highlighted in yellow.

6. Further elaborate on the specific advantages or innovations of this method.

Author’s response: It has been answered in the same description where comparison with other scholars’ work has been made, as stated above.

Reviewer #2: 

1. The introduction of this submission should be improved further. The key differences between your work and previous studies should be clarified. A point-to-point way is recommended to state the main contributions.

Author’s response: The paper has been reviewed by the author after reviewers’ comments and the content has been improved. Hope it will satisfy the reviewer.

2. PV modelling section is very traditional, and this reviewer suggests this section can be shortened properly.

Author’s response: Author appreciate reviewer’s comment on PV modelling. In response, it is stated that PV modelling contains equations for modelling the related text to describe the terms used in model equations. No extra content has been added and author has aimed to contain modelling section in as small portion as possible.

3. The writing style of this work likes a technical report, but not a scientific paper.

Author’s response: Paper has been revised with minor changes as per reviewers’ comments. Hopefully the new draft resembles more like paper.

 4. A comparison of your method and other work should be carried out.

Author’s response: Done at the end of Results and Discussion chapter.

 5. Most of the figures have poor quality, such as, lacking unit.

Author’s response: This issue has been addressed in the new draft, where care has been taken while generating new figures for the paper.

---

## [Decision Letter · Decision Letter 1]

17 Nov 2021

A model-based approach for detecting and identifying faults on the D.C. side of a P.V. system using electrical signatures from I-V characteristics

PONE-D-21-20995R1

Dear Dr. Khan,

We’re pleased to inform you that your manuscript has been judged scientifically suitable for publication and will be formally accepted for publication once it meets all outstanding technical requirements.

Kind regards,

Lei Chen, Ph.D.

Academic Editor

PLOS ONE

Additional Editor Comments (optional):

Reviewers' comments:

Reviewer's Responses to Questions

**Comments to the Author**

1. If the authors have adequately addressed your comments raised in a previous round of review and you feel that this manuscript is now acceptable for publication, you may indicate that here to bypass the “Comments to the Author” section, enter your conflict of interest statement in the “Confidential to Editor” section, and submit your "Accept" recommendation.

Reviewer #1: All comments have been addressed

Reviewer #2: All comments have been addressed

2. Is the manuscript technically sound, and do the data support the conclusions?

Reviewer #1: Yes

Reviewer #2: Yes

3. Has the statistical analysis been performed appropriately and rigorously? 

Reviewer #1: Yes

Reviewer #2: Yes

4. Have the authors made all data underlying the findings in their manuscript fully available?

Reviewer #1: Yes

Reviewer #2: Yes

5. Is the manuscript presented in an intelligible fashion and written in standard English?

Reviewer #1: Yes

Reviewer #2: Yes

6. Review Comments to the Author

Reviewer #1: (No Response)

Reviewer #2: (No Response)

7. PLOS authors have the option to publish the peer review history of their article (what does this mean?). If published, this will include your full peer review and any attached files.

Reviewer #1: No

Reviewer #2: No

---

## [Editor Report · Acceptance letter]

23 Dec 2021

PONE-D-21-20995R1 

A model-based approach for detecting and identifying faults on the D.C. side of a P.V. system using electrical signatures from I-V characteristics 

Dear Dr. Khan:

I'm pleased to inform you that your manuscript has been deemed suitable for publication in PLOS ONE. Congratulations! Your manuscript is now with our production department. 

Kind regards, 

on behalf of

Professor Lei Chen 

Academic Editor

PLOS ONE